# On the Implicit Biases of Architecture & Gradient Descent

## Abstract

Do neural networks generalise because of bias in the functions returned by gradient descent, or bias already present in the network architecture?

*¿Por qué no los dos?*

This paper finds that while typical networks that fit the training data already generalise fairly well, gradient descent can further improve generalisation by selecting networks with a large margin. This conclusion is based on a careful study of the behaviour of infinite width networks trained by Bayesian inference and finite width networks trained by gradient descent. To measure the implicit bias of architecture, new technical tools are developed to both *analytically bound* and *consistently estimate* the average test error of the neural network–Gaussian process (NNGP) posterior. This error is found to be already better than chance, corroborating the findings of Valle-Pérez et al. (2019) and underscoring the importance of architecture. Going beyond this result, this paper finds that test performance can be substantially improved by selecting a function with much larger margin than is typical under the NNGP posterior. This highlights a curious fact: *minimum a posteriori* functions can generalise best, and gradient descent can select for those functions. In summary, new technical tools suggest a nuanced portrait of generalisation involving both the implicit biases of architecture and gradient descent.

## 1 Introduction

Following an influential paper by Zhang et al. (2017), the basic question of *why neural networks generalise* is generally regarded to be open. The authors demonstrate a surprising fact: deep learning generalises even when the neural network is expressive enough to represent functions that do not generalise. In turn, this implies that the theory of Vapnik & Chervonenkis (1971)—based on uniform convergence of train error to population error—does not explain generalisation in neural networks.

Several hypotheses have since been proposed to fill this theoretical vacuum. One prominent hypothesis states that, while most neural network solutions do not generalise well, there is an implicit bias in the kinds of functions returned by gradient descent (Soudry et al., 2018). In sharp contrast, a second hypothesis states that the solution space of a neural network is dominated by simple functions, while the complex kinds of functions that overfit are relatively rare (Valle-Pérez et al., 2019).

This latter hypothesis dovetails with a particular "PAC-Bayesian" theorem of McAllester (1998), which bounds the average population error of all classifiers consistent with a training sample. If this bound is small, then the complex functions that overfit must indeed be rare. In more technical terms, the PAC-Bayesian theorem can provide a meaningful certificate of generalisation even for machine learning models with an infinite Vapnik-Chervonenkis dimension, or an arbitrarily large number of parameters, by properly accounting for the measure of those very complex functions.

This paper takes a more nuanced position between these two hypotheses. While the *average* population error of all neural networks that fit a training sample is found to be good (and *certifiably good* by the PAC-Bayesian theorem), it is still possible for certain networks with special properties to perform substantially *better than average* (and likewise substantially better than the PAC-Bayes bound). Moreover, gradient descent may be used to specifically target these special networks. This subtly counters a position put forward by Mingard et al. (2021), which suggests that gradient descent may be well-modelled as sampling randomly from a particular Bayesian posterior distribution.

To support these claims, a careful study of the behaviour of both infinite width and finite width neural networks is conducted. In particular, this paper makes the following technical contributions:

Section 3    *Implicit Bias of Architecture.* A purely analytical PAC-Bayes bound (Theorem 2) on the population error of the neural network–Gaussian process (NNGP) posterior for binary classification is derived. The bound furnishes an interpretable measure of model complexity that depends on both the architecture and the training data. This exact analytical bound improves upon an approximate computational approach due to Valle-Pérez et al. (2019).

Section 4    *Testing the Bound.* The new bound is found to be both non-vacuous and correlated with the test error of finite width multilayer perceptrons trained by gradient descent. This provides supporting evidence for the important role of architecture in generalisation, as put forward by Valle-Pérez et al. (2019). Still, a gap exists with the performance of gradient descent.

Section 5    *Implicit Bias of Gradient Descent.* Going further beyond the work of Valle-Pérez et al. (2019), a new theoretical tool (Theorem 3) is developed to enable consistent estimation of the average error of the NNGP posterior on a given holdout set. The average is found to be significantly worse than the holdout performance of Gaussian process draws with large margin. The experiment is repeated for finite width neural networks trained by gradient descent, and the same qualitative phenomenon persists. This finding demonstrates the ability of gradient descent to select large margin functions with vanishing posterior probability that nonetheless generalise significantly better than the posterior average.

## 2    RELATED WORK

**Margin-based generalisation theory**    A rich body of work explores the connection between margin and generalisation. For instance, Bartlett et al. (2017) propose a margin-based complexity measure for neural networks derived via Rademacher complexity analysis, and Neyshabur et al. (2018) derive a similar result via PAC-Bayes analysis. Neyshabur et al. (2019) test these bounds experimentally, finding them to be vacuous and to scale poorly with network width. Biggs & Guedj (2021) further develop this style of bound. Margin-based PAC-Bayes bounds go back at least to the work of Herbrich (2001), who derived such a bound for linear classification. The standard idea is that a solution with large margin implies the existence of nearby solutions with the same training error (but perhaps smaller margin), facilitating a more spread out PAC-Bayes posterior. This style of margin-based PAC-Bayes bound does not provide guidance on whether the original large margin classifier should generalise better or worse than the additional nearby classifiers included in the posterior.

**Non-vacuous bounds for neural networks**    Aside from PAC-Bayes, many styles of generalisation bound for neural networks are vacuous (Dziugaite & Roy, 2017). Even many PAC-Bayes bounds are vacuous (Achille & Soatto, 2018; Foret et al., 2021) due to their choice of PAC-Bayes prior. Dziugaite & Roy (2017) construct a *non-vacuous* weight space PAC-Bayes bound by iteratively optimising both the PAC-Bayes prior and posterior by gradient descent. Wu et al. (2020) extend this approach to involve Hessians. Meanwhile, Valle-Pérez et al. (2019) instantiate a non-vacuous PAC-Bayes bound in the function space of neural networks via the NNGP correspondence (Neal, 1994; Lee et al., 2018). The evaluation of this bound involves an iterative and statistically inconsistent expectation-propagation approximation. Unlike this paper's purely analytical bound in Theorem 2, these non-vacuous bounds are all evaluated by iterative computational procedures.

**Implicit bias of gradient descent**    Much research has gone into the role that gradient descent plays in neural network generalisation. For instance, Zhang et al. (2017) suggest that gradient descent may converge to solutions with special generalisation properties, while Wilson et al. (2017) suggest that gradient descent may have a more favourable implicit bias than the Adam optimiser. Azizan et al. (2021) make a related study in the context of mirror descent. Keskar et al. (2017) point to an implicit bias present in mini-batch stochastic gradient descent, but Geiping et al. (2021) argue against the importance of stochasticity. Meanwhile, Soudry et al. (2018) observe how gradient descent combined with certain loss functions converges to max-margin solutions; Wei et al. (2018) make a similar observation. On a different tack, Mingard et al. (2021), Valle-Pérez & Louis (2020) and Valle-Pérez et al. (2019) argue that the implicit bias of neural architecture is more important than that of gradient descent for understanding generalisation, and so long as gradient descent does not select solutions pathologically, generalisation should occur.

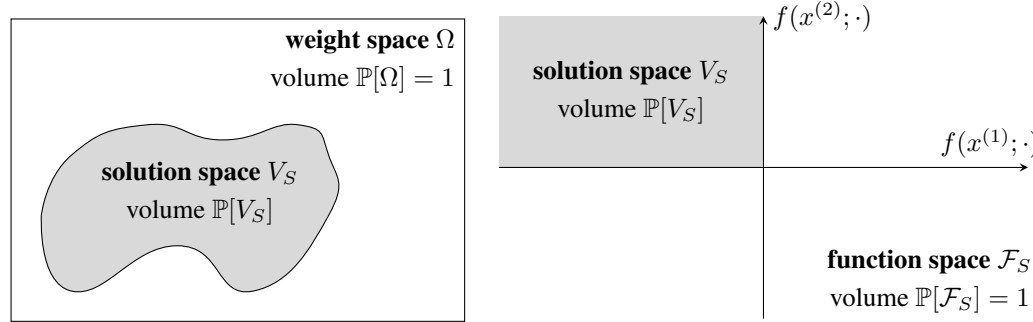

Figure 1: The solution space $V_S$ of a learning task denotes the subset of classifiers that attain zero error on a dataset $S$. While the solution space can have a complicated geometry when described in weight space, in function space the solution space of a binary classification task is just an orthant.

## 3 IMPLICIT BIAS OF ARCHITECTURE

This section derives an analytical bound on the population error of an infinitely wide neural network averaged over all weight configurations that attain 0% train error—in other words, the average population error of the NNGP posterior. Since this quantity assesses the performance of all solutions rather than just those returned by gradient descent, it is intended to measure the implicit bias of architecture. In Section 4, the bound is found to be non-vacuous for multilayer perceptrons. The source of a substantial gap with the results of gradient descent training is investigated in Section 5, and ultimately attributed to an additional implicit bias of gradient descent: namely *margin*.

### 3.1 PRELIMINARY NOTATION

Consider a training dataset $S$ of $n$ input–label pairs: $S = \{(x^{(1)}, y^{(1)}), ..., (x^{(n)}, y^{(n)})\}$. The inputs $x^{(i)} \in \mathbb{R}^{d_0}$ are embedded in Euclidean $d_0$-space, while to simplify the analysis the paper shall restrict to binary labels $y^{(i)} \in \{\pm 1\}$. A classifier $f$ shall depend on a weight vector $w \in \Omega$, where $\Omega$ denotes the *weight space*. In particular, for an input $x$ the prediction is given by $\text{sign} f(x; w)$.

One is interested in the relationship between the *train error* $\varepsilon_S$ of a classifier on a dataset $S$ to the *population error* $\varepsilon_D$ on a data distribution $D$. For a weight vector $w \in \Omega$, these are defined as:

$$\varepsilon_S(w) := \frac{1}{n} \sum_{i=1}^{n} \mathbb{I} \left[ \text{sign} f(x^{(i)}, w) \neq y^{(i)} \right]; \qquad \varepsilon_D(w) := \mathbb{P}_{(x,y) \sim D} \left[ \text{sign} f(x; w) \neq y \right]. \quad (1)$$

The *solution space* $V_S := \{ w \in \Omega \,|\, \varepsilon_S(w) = 0 \}$ denotes those classifiers that attain zero train error.

### 3.2 GEOMETRY OF SOLUTIONS IN FUNCTION SPACE

The function space $\mathcal{F}_S$ is defined to be the set of outputs that the classifier $f$ can realise on dataset $S$:

$$\mathcal{F}_S := \left\{ \left( f(x^{(1)}, w), ..., f(x^{(n)}, w) \right) \,\middle|\, w \in \Omega \right\} \subset \mathbb{R}^n. \quad (2)$$

In weight space the geometry of solutions can be arbitrarily complicated, but in function space the solutions occupy the single orthant picked out by the binary training labels.

To measure the volume of the solution space, one can define a measure $\mathbb{P}$ on weight space. It is convenient to enforce $\mathbb{P}[\Omega] = 1$ so that $\mathbb{P}$ is a probability measure. In the absence of a better alternative, $\mathbb{P}$ is usually set to a multivariate Gaussian or uniform distribution. The measure $\mathbb{P}$ on weight space induces a measure $\mathbb{P}$ on function space via the relation:

$$\mathbb{P}[F \subset \mathcal{F}_S] := \mathbb{P} \left\{ w \in \Omega \,\middle|\, \left( f(x^{(1)}, w), ..., f(x^{(n)}, w) \right) \in F \right\}. \quad (3)$$

The volume of solutions is denoted $\mathbb{P}[V_S]$, and can be computed either in weight or function space. In function space, $\mathbb{P}[V_S]$ is an orthant probability. The situation is visualised in Figure 1.

### 3.3 PAC-BAYES THEORY

PAC-Bayes relates the volume of solutions to their average population error. The following result was derived by Valle-Pérez et al. (2019) as a corollary of a theorem due to Langford & Seeger (2001). The result is similar in form to a theorem of McAllester (1998).

**Theorem 1** (A PAC-Bayesian theorem). *First, fix a probability measure $\mathbb{P}$ over the weight space $\Omega$ of a classifier. Let $S$ denote a training set of $n$ datapoints sampled iid from the data distribution $D$ and let $V_S$ denote the corresponding solution space. Consider the population error $0 \le \varepsilon_D(w) \le 1$ of weight setting $w \in \Omega$, and its average over the solution space $\varepsilon_D(V_S) := \mathbb{E}_{w\sim\mathbb{P}}[\varepsilon_D(w) \,|\, w \in V_S]$. Then, for a proportion $1 - \delta$ of draws of the training set $S$,*

$$\varepsilon_D(V_S) \le \ln \frac{1}{1 - \varepsilon_D(V_S)} \le \frac{\ln \frac{1}{\mathbb{P}[V_S]} + \ln \frac{2n}{\delta}}{n - 1}. \tag{4}$$

For large $n$, the $\ln 2n/\delta$ term is negligible and the result says that the population error averaged over solutions is less than the ratio of the negative log volume of solutions to the number of training points.

### 3.4 VOLUME OF SOLUTIONS VIA GAUSSIAN ORTHANT PROBABILITIES

Since infinitely wide neural networks induce a Gaussian measure on function space, computing the volume of the solution space $\mathbb{P}[V_S]$ amounts to computing a Gaussian orthant probability.

In more detail, let $\mathbb{P}$ denote a measure on the weight space $\Omega$ of an infinitely wide neural network satisfying the conditions of the NNGP correspondence—see Theorem 4 in Appendix A.3 for an example. Then for a weight vector $w \sim \mathbb{P}$, the network outputs on a training set $S$ are distributed:

$$f(x^{(1)}; w), ..., f(x^{(n)}; w) \sim \mathcal{N}(0, \Sigma), \tag{5}$$

with covariance $\Sigma_{ij} := \mathbb{E}_{w\sim\mathbb{P}}[f(x^{(i)}; w) f(x^{(j)}; w)]$.

Therefore, under the NNGP correspondence, the volume of solutions computed in function space is just the Gaussian probability of the orthant picked out by the binary training labels:

$$\mathbb{P}[V_S] = \mathbb{P}_{\varphi\sim\mathcal{N}(0,\Sigma)}[\operatorname{sign}\varphi_1 = y^{(1)}, ..., \operatorname{sign}\varphi_n = y^{(n)}]. \tag{6}$$

To facilitate estimating and bounding this probability, this paper has derived the following lemma.

**Lemma 1** (Gaussian orthant probability). *For a covariance matrix $\Sigma \in \mathbb{R}^{n\times n}$, and a binary vector $y \in \{\pm 1\}^n$, let $p$ denote the corresponding Gaussian orthant probability:*

$$p := \mathbb{P}_{\varphi\sim\mathcal{N}(0,\Sigma)}[\operatorname{sign}(\varphi) = y]. \tag{7}$$

*Letting $\mathbb{I}$ denote the $n \times n$ identity matrix, $\odot$ the elementwise product and $|\cdot|$ the elementwise absolute value, then $p$ may be equivalently expressed as:*

$$p = \frac{1}{2^n} \mathbb{E}_{u\sim\mathcal{N}(0,\mathbb{I})} \left[ e^{-\frac{1}{2}(y\odot|u|)^T \left( \sqrt[n]{\det\Sigma}\,\Sigma^{-1} - \mathbb{I} \right)(y\odot|u|)} \right] =: e^{-\mathcal{C}_0(\Sigma, y)}, \tag{8}$$

*and $p$ may be bounded as follows:*

$$p \ge \frac{1}{2^n} e^{\frac{n}{2} - \sqrt[n]{\det\Sigma}\left[ (\frac{1}{2} - \frac{1}{\pi})\operatorname{tr}(\Sigma^{-1}) + \frac{1}{\pi} y^T\Sigma^{-1}y \right]} =: \frac{1}{2^n} e^{\frac{n}{2} - \mathcal{C}_1(\Sigma, y)}. \tag{9}$$

The proof is given in Appendix A.2. Equation 8 yields an unbiased Monte Carlo estimator of Gaussian orthant probabilities, and Inequality 9 yields a lower bound. The complexity measures $\mathcal{C}_0$ and $\mathcal{C}_1$ are defined for later use.

To gain intuition about the lemma, observe that $1/2^n$ is the orthant probability for an isotropic Gaussian. Depending on the degree of *anisotropy* $\sqrt[n]{\det\Sigma\,\Sigma^{-1}} - \mathbb{I}$ inherent in the covariance matrix $\Sigma$, Equation 8 captures how the orthant probability may either be exponentially amplified or suppressed compared to $1/2^n$.

As an aside, using Inequality 9 to lower bound Equation 6 has a Bayesian interpretation. Since the volume of solutions may be written $\mathbb{P}[V_S] = \int_\Omega d\mathbb{P}(w)\,\mathbb{I}[w \in V_S]$, it may be interpreted as the *Bayesian evidence* for the network architecture under the *likelihood function* $\mathbb{I}[w \in V_S]$. A Bayesian would then refer to Inequality 9 as an *evidence lower bound*.

The following generalisation bound is a basic consequence of Theorem 1, Equation 6 and Lemma 1:

**Theorem 2** (Upper bound on the average population error of an infinitely wide neural network).
*First, fix a probability measure $\mathbb{P}$ over the weight space $\Omega$ of an infinitely wide neural network. Let $S$
denote a training set of $n$ datapoints sampled iid from the data distribution $D$, let $y \in \{\pm 1\}^n$ denote
the binary vector of training labels, and let $V_S$ denote the corresponding solution space. Consider the
population error $0 \leq \varepsilon_D(w) \leq 1$ of weight setting $w \in \Omega$, and its average over the solution space
$\varepsilon_D(V_S) := \mathbb{E}_{w \sim \mathbb{P}}[\varepsilon_D(w) \,|\, w \in V_S]$. Let $\Sigma$ denote the NNGP covariance matrix (Equation 5). Then,
for a proportion $1 - \delta$ of draws of the training set $S$,*

$$\varepsilon_D(V_S) \leq \ln \frac{1}{1 - \varepsilon_D(V_S)} \leq \frac{\mathcal{C}_0(\Sigma, y) + \ln \frac{2n}{\delta}}{n - 1} \leq \frac{\frac{n}{5} + \mathcal{C}_1(\Sigma, y) + \ln \frac{2n}{\delta}}{n - 1}, \tag{10}$$

*where the complexity measures $\mathcal{C}_0$ and $\mathcal{C}_1$ are defined in Lemma 1.*

Since the complexity measure $\mathcal{C}_1$ is an analytical function of the NNGP covariance matrix $\Sigma$ and the
binary vector of training labels $y$, Theorem 2 is an analytical generalisation bound for the NNGP
posterior. For large $n$, the result simplifies to: $\varepsilon_D(V_S) \lesssim \mathcal{C}_0(\Sigma, y)/n \leq 1/5 + \mathcal{C}_1(\Sigma, y)/n$. So the
bound depends on the ratio of complexity measures $\mathcal{C}_0$ and $\mathcal{C}_1$ to the number of datapoints $n$.

To gain further intuition about Theorem 2, consider two special cases. First, suppose that the neural
architecture induces no correlation between any pair of distinct data points, such that $\Sigma = \mathbb{I}$. Then
$\varepsilon_D(V_S) \lesssim \mathcal{C}_0(\mathbb{I}, y)/n = \ln 2 \approx 0.7$ and the bound is worse than chance. This corresponds to
*pure memorisation* of the training labels. Next, suppose that the neural architecture induces strong
intra-class correlations and strong inter-class anti-correlations, such that $\Sigma_{ij} = y_i y_j$. Although this $\Sigma$
is singular, it may be seen directly that $\mathbb{P}[V_S] = 1/2$. Then by Theorem 1, $\varepsilon_D(V_S) \lesssim \ln 2/n$ which
is much better than chance for large $n$. This corresponds to *pure generalisation* from the training
labels. Interpolating between these two extremes would suggest that a good neural architecture would
impose a prior on functions with strong intra-class and weak inter-class correlations.

## 4 TESTING THE BOUND

This section compares the generalisation bound for infinite width networks (Theorem 2) to the
performance of finite width multilayer perceptrons (MLPs) trained by gradient descent. The bound is
found to be non-vacuous and correlated with the effects of varying depth and dataset complexity. Still,
there is a substantial gap between the bound and gradient descent, which is investigated in Section 5.

Three modified versions of the MNIST handwritten digit dataset (LeCun et al., 1998) of varying
"hardness" were used in the experiments, as detailed in Figure 2 (top left). MLPs were trained with $L$
layers and $W$ hidden units per layer. Specifically, each $28\text{px} \times 28\text{px}$ input image $x$ was flattened
to lie in $\mathbb{R}^{784}$, and normalised to satisfy $\|x\|_2 = \sqrt{784}$. The networks consisted of an input layer
in $\mathbb{R}^{784 \times W}$, $(L - 2)$ layers in $\mathbb{R}^{W \times W}$, and an output layer in $\mathbb{R}^{W \times 1}$. The nonlinearity $\varphi$ was set
to $\varphi(z) := \sqrt{2} \cdot \max(0, z)$. For this architecture, the width $W \to \infty$ kernel is the *compositional
arccosine kernel* described in Theorem 4 in Appendix A.3. For the finite width networks, the training
loss was set to square loss using the binary training labels as regression targets, and the networks
were trained for 100 epochs using the Nero optimiser (Liu et al., 2021) with an initial learning rate
of 0.01 decayed by a factor of 0.9 every epoch, and a mini-batch size of $\min(50, \text{training set size})$
data points. The final train error was 0% in all reported experiments. All bounds were computed
with a failure probability of $\delta = 0.01$, and $\mathcal{C}_0$ was estimated using $10^6$ Monte-Carlo samples. All
experiments were run on one NVIDIA Titan RTX GPU.

The generalisation bound of Theorem 2 was first compared across three datasets of varying complexity.
The network architecture was set to a depth $L = 7$ MLP, and the bound was computed via Monte-
Carlo estimation of $\mathcal{C}_0$. The results are shown in Figure 2 (top right). The bound was found to reflect
the relative hardness of the datasets. For *random labels*, the bound was vacuous as desired. Next,
the generalisation bound was compared against the empirical performance of a depth $L = 7$ MLP
trained on the *decimal digits* dataset. The results are shown in Figure 2 (bottom left). While loose
compared to the holdout error of the finite width network, the bound is still non-vacuous. Finally,
the effect of varying network depth was investigated on the *decimal digits* dataset. Two depths
were compared: $L = 2$ and $L = 7$. The results are shown in Figure 2 (bottom right). The bounds
(computed via Monte-Carlo estimation of $\mathcal{C}_0$) appear to predict the relative holdout performance of
the two architectures at finite width. These results corroborate those of Valle-Pérez & Louis (2020)
but without the use of the statistically inconsistent expectation-propagation approximation.

| MNIST Variant | Inputs | Labels |
|---|---|---|
| random labels | $\{0, 1, ..., 9\}$ | coin flip |
| decimal digits | $\{0, 1, ..., 9\}$ | parity |
| binary digits | $\{0, 1\}$ | parity |

Figure 2: Testing Theorem 2—the bound on the average population error of the NNGP posterior. The bound is found to be non-vacuous and correlated with the effects of varying network depth and dataset complexity. For all curves, the mean and range are plotted over three global random seeds. A substantial gap is visible between the bound and the results of gradient descent training, which is investigated in Section 5.

Top left: The three datasets used in the experiments, listed in order of hardness. For *random labels* there is no meaningful relationship between image and label, so generalisation is impossible. *Binary digits* is easier than *decimal digits* because each class is less diverse.

Top right: Comparing generalisation bounds from Theorem 2 for datasets of varying hardness. The curves are computed by Monte-Carlo estimation of $\mathcal{C}_0$ for an infinite width depth 7 MLP. The ordering of the bounds reflects the dataset hardness. For *random labels* the bound is rightfully vacuous.

Bottom left: Comparing generalisation bounds from Theorem 2 to the results of training finite width ($W = 5000$ and $W = 10000$) networks by gradient descent. The bounds are computed by both Monte-Carlo estimation of $\mathcal{C}_0$ (referred to as $W = \infty$ estimator) and exact computation of $\mathcal{C}_1$ (referred to as $W = \infty$ bound). The comparison is made for depth 7 MLPs on the *decimal digits* dataset. The exact bound computed via $\mathcal{C}_1$ is looser than the $\mathcal{C}_0$ bound computed via Monte-Carlo estimation but only slightly, and both are non-vacuous above 150 datapoints. While the bounds follow the same trend as the results of training finite width networks, a substantial gap is visible.

Bottom right: Comparing generalisation bounds from Theorem 2 to the results of training finite width networks by gradient descent on *decimal digits*, for networks of varying depth. The bounds are computed by exact computation of $\mathcal{C}_1$. The ordering of the bounds matches the finite width results, but again a substantial gap is visible between the bounds and the results of gradient descent training.

## 5 Implicit Bias of Gradient Descent

In Section 3, a bound was derived on the average population error of all infinitely wide neural networks that fit a certain training set. Since the bounded quantity measures the performance of all solutions rather than just those returned by gradient descent, it is intended to assess the implicit bias of architecture. In Section 4, this bound was tested and found to be non-vacuous. Still a substantial gap was found between the bound and the performance of finite width networks trained by gradient descent. This gap could arise for several potential reasons:

    i) slackness in the bounding technique;

    ii) a difference between infinite width and finite width neural networks;

    iii) an additional implicit bias of gradient descent.

This section tests these various possibilities, ultimately concluding that gradient descent does have an important additional implicit bias: the ability to control the margin of the returned network.

### 5.1 An Exact Formula for the Average Holdout Error of Solutions

To investigate whether slackness in the bounding technique is responsible for the gap, an additional theoretical tool was developed: a formula for the holdout error of a binary classifier averaged over the solution space. In contrast to Theorem 2 which gives an upper bound on population error, Theorem 3 is an equality and therefore does not suffer from slackness. The proof is given in Appendix A.1.

**Theorem 3** (Average holdout error of a binary classifier). *First, fix a probability measure $\mathbb{P}$ over the weight space $\Omega$ of a binary classifier. Let $S$ denote a training set and let $V_S$ denote the solution space. For a holdout set $T$ of $m$ datapoints, consider the holdout error $0 \leq \varepsilon_T(w) \leq 1$ of weight setting $w \in \Omega$, and its average over the solution space $\varepsilon_T(V_S) := \mathbb{E}_{w \sim \mathbb{P}}[\varepsilon_T(w) \,|\, w \in V_S]$. Then:*

$$\varepsilon_T(V_S) = \frac{1}{m} \sum_{(x,y) \in T} \frac{\mathbb{P}[V_{S \cup (x,-y)}]}{\mathbb{P}[V_S]}. \tag{11}$$

In words, the average holdout error over solutions equals the reduction in the volume of solutions when the training set is augmented with a negated holdout point, averaged over the holdout set. For an infinitely wide neural network, Equation 11 involves computing a sum of ratios of Gaussian orthant probabilities. These probabilities can be consistently estimated by Equation 8 in Lemma 1.

### 5.2 Three-Way Comparison of Holdout Error

Armed with Theorem 3, this subsection makes a three-way comparison between the holdout error of infinite width networks averaged over solutions, the holdout error of finite width networks averaged over solutions, and the holdout error of finite width networks trained by gradient descent.

To compute the holdout error of finite width networks averaged over solutions, weight vectors were randomly sampled and all non-solutions were discarded. To make this process computationally tractable, a very small training set was used consisting of only 5 samples from *binary digits*. A holdout set of 50 datapoints was used, and the network was set to a 7-layer MLP. The results were:

| | |
|---|---|
| average holdout error at infinite width: | $0.337 \pm 0.001$; |
| average holdout error at width 10,000: | $0.33 \pm 0.01$; |
| gradient descent holdout error at width 10,000: | $0.178 \pm 0.007$. |

Based on these results, three comments are in order. First, the close agreement between the average holdout error at finite and infinite width suggests that the infinite width limit may not be responsible for the significant gap observed in Section 4. Second, the significant gap between the gradient descent holdout error and the holdout error averaged over solutions suggests slackness in the bounding technique in Theorem 2 may also not be the main culprit. Third, the significant gap between the gradient descent holdout error and the holdout error averaged over solutions suggests that gradient descent *does* have an extra implicit bias. The next subsection attempts to diagnose this implicit bias.

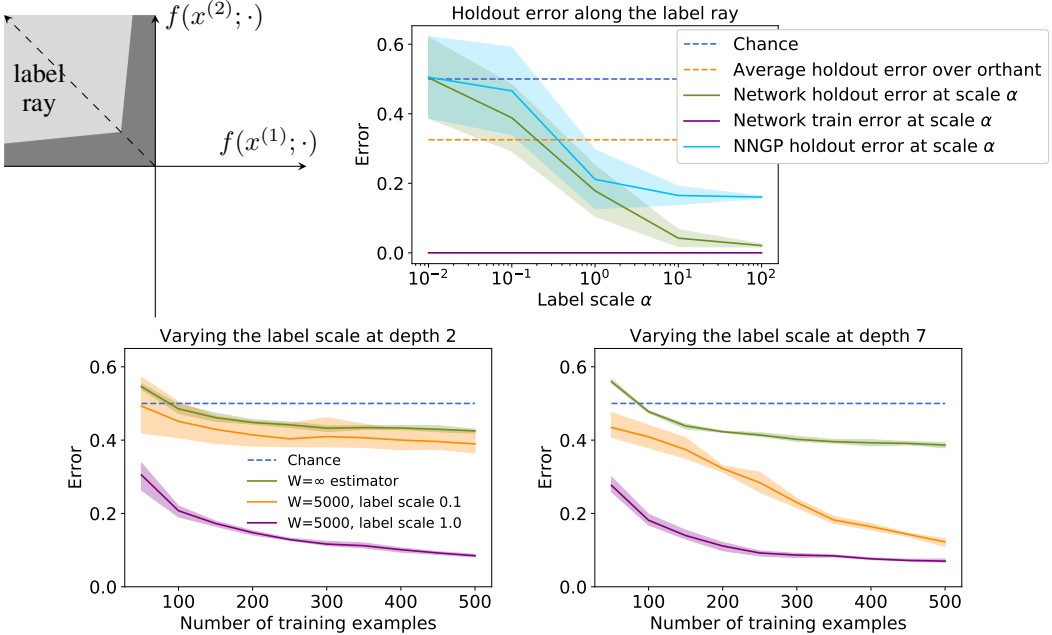

Figure 3: Exploring generalisation performance along the *label ray* defined in Section 5.3. Functions further along the ray are found to generalise substantially better than average, for both NNGP draws and finite width networks trained by gradient descent.

Top left: Schematic diagram illustrating the label ray in function space. While all functions in the grey shaded orthant attain zero train error, the darker shaded solutions have small margin and thus—under a Gaussian process model—holdout predictions are expected to be driven by random fluctuations.

Top right: Holdout error along the label ray of the small learning task described in Section 5.2. Results are shown for both networks trained by gradient descent and NNGP posterior samples. The label scale $\alpha$ is defined in Section 5.3 and measures the distance of the function along the label ray. NNGP posterior samples were generated by sampling from a Gaussian distribution with mean and covariance given by Equations 12 and 13. Networks trained by gradient descent were found by training with the loss function given in Equation 14. Shading shows the standard deviation over 100 random intialisations or posterior samples. Deep into function space (large $\alpha$) the holdout error is significantly better than for functions close to the origin (small $\alpha$). The average holdout error over the orthant, as computed in Section 5.2, is also plotted. Large margin networks and NNGP posterior samples both significantly outperform the orthant average. Finally, a gap is visible between large margin NNGP posterior samples and large margin networks trained by gradient descent—this may be due to an additional and undiagnosed implicit bias of gradient descent.

Bottom: Repeating experiments from Section 4 with a smaller label scale to sanity check the findings. For a depth 2 and a depth 7 MLP on the *decimal digits* dataset, the experiment from Section 4 was repeated with the loss function set to $\mathcal{L}_{\alpha=0.1}$ as defined in Equation 14. The infinite width $W = \infty$ curve shows the Theorem 2 bound estimated via $\mathcal{C}_0$. The other two curves show the results of networks trained by gradient descent with $\mathcal{L}_{\alpha=0.1}$ and $\mathcal{L}_{\alpha=1}$. Despite all networks attaining 0% train error, holdout error was significantly worse for networks trained using $\mathcal{L}_{\alpha=0.1}$. Also, the networks trained using $\mathcal{L}_{\alpha=0.1}$ exhibit a substantially smaller gap with the Theorem 2 upper bound.

### 5.3 Diagnosis: Margin

This subsection finds that gradient descent has an important implicit bias in determining the margin of the returned network. This conclusion is made by studying the generalisation error of functions along the *label ray* in function space—depicted in Figure 3 (top left). For a training set $S$ with a vector of binary labels $y \in \{\pm 1\}^n$, a function $\alpha$-*far* along the label ray refers to the point $\alpha y \in \mathcal{F}_S$.

For the case of a zero mean NNGP, the predictive distribution on a holdout set $T$ conditioned on training labels $\alpha$-far along the label ray is given by Bishop (2006, Chapter 2.3.1):

$$\text{posterior mean} = \alpha \times \Sigma_{TS} \Sigma_{SS}^{-1} y, \tag{12}$$

$$\text{posterior covariance} = \Sigma_{TT} - \Sigma_{TS} \Sigma_{SS}^{-1} \Sigma_{ST}, \tag{13}$$

where $\Sigma_{TS}$ is the covariance between holdout and train inputs, and $\Sigma_{ST}$, $\Sigma_{SS}$ and $\Sigma_{TT}$ are defined analogously. For large $\alpha$, predictions are driven by the posterior mean, whereas for small $\alpha$, they are driven by random fluctuations. So one expects that letting $\alpha \to \infty$ should improve holdout error.

For the case of gradient descent training, functions $\alpha$-far along the label ray can be returned by minimising the following loss function:

$$\mathcal{L}_\alpha(W) := \frac{1}{n} \sum_{i=1}^n \left( f(x^{(i)}; W) - \alpha y^{(i)} \right)^2. \tag{14}$$

A subtle but important point is that for gradient descent training with Nero (Liu et al., 2021) the norms of the weight matrices are constrained, thus $\alpha$ controls a properly normalised notion of margin.

An experiment was performed to measure the holdout error of networks $\alpha$-far along the label ray—for both NNGP draws and neural networks selected by Nero—for $\alpha$ ranging from $10^{-2}$ to $10^2$. As can be seen in Figure 3 (top right), varying $\alpha$ appears to directly control the holdout error, despite all solutions attaining $0\%$ train error. Moreover, large $\alpha$ solutions significantly outperform the average holdout error over the solution space. This suggests that gradient descent possesses a significant implicit bias in its ability to control margin—going beyond the implicit bias of architecture.

To sanity check this result, experiments from Section 4 were repeated with the loss function $\mathcal{L}_{\alpha=0.1}$ replacing $\mathcal{L}_{\alpha=1}$. As can be seen in Figure 3 (bottom), the holdout performance was significantly diminshed at $\alpha = 0.1$, as was the gap with the PAC-Bayes bound from Theorem 2.

## 6 Discussion and Conclusion

This paper has explored the separate implicit biases of architecture and gradient descent. Section 3 derived an analytical generalisation bound on the NNGP posterior, Section 4 found this bound to be non-vacuous, while Section 5 showed that large margin functions substantially outperform the bound.

The findings in this paper support the importance of the *implicit bias of architecture*: reproducing the findings in Valle-Pérez & Louis (2020) but with improved technical tools, the average generalisation performance of architecture was already found to be good. But the *implicit bias of gradient descent* was also found to be important. In particular, in contrast to an assumption made by Valle-Pérez & Louis (2020) that gradient descent "samples the zero-error region close to uniformly", it was found that gradient descent can be used to target *zero-error functions with large margin*. This also subtly counters a proposal by Mingard et al. (2021) that gradient descent acts like a "Bayesian sampler". In particular, gradient descent can target functions with margin $\alpha \to \infty$ for which the Bayesian posterior probability $\to 0$. And indeed, these *minimum a posteriori* functions seem to generalise best.

One direction of future work suggested by these results is an improvement to the function space PAC-Bayes theory to account for margin. While margin-based PAC-Bayes bounds do already exist (Herbrich, 2001; Langford & Shawe-Taylor, 2003; Neyshabur et al., 2018), these bounds operate in weight space and further do not seem to imply an advantage to selecting the max-margin classifier over the other classifiers included in the PAC-Bayes posterior.

Ultimately, a generalisation theory that properly accounts for the various implicit biases of deep learning could provide a more principled basis for both neural architecture design as well as the design of new regularisation schemes. It is hoped that the new analytical results as well as experimental insights included in this paper contribute a step towards reaching that goal.

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

## APPENDIX A   PROOFS

### A.1   FUNCTION SPACE ORTHANT ARITHMETIC

The following PAC-Bayesian theorem is due to Valle-Pérez et al. (2019):

**Theorem 1** (A PAC-Bayesian theorem). *First, fix a probability measure $\mathbb{P}$ over the weight space $\Omega$ of a classifier. Let $S$ denote a training set of $n$ datapoints sampled iid from the data distribution $D$ and let $V_S$ denote the corresponding solution space. Consider the population error $0 \leq \varepsilon_D(w) \leq 1$ of weight setting $w \in \Omega$, and its average over the solution space $\varepsilon_D(V_S) := \mathbb{E}_{w \sim \mathbb{P}}[\varepsilon_D(w) \,|\, w \in V_S]$. Then, for a proportion $1 - \delta$ of draws of the training set $S$,*

$$\varepsilon_D(V_S) \leq \ln \frac{1}{1 - \varepsilon_D(V_S)} \leq \frac{\ln \frac{1}{\mathbb{P}[V_S]} + \ln \frac{2n}{\delta}}{n - 1}. \tag{4}$$

*Proof.* The first inequality is a basic property of logarithms. The second inequality follows from Theorem 3 of Langford & Seeger (2001), by setting the prior measure to $P(\cdot) = \mathbb{P}(\cdot)$ and the posterior measure to the conditional $Q(\cdot) = \mathbb{P}(\cdot|V_S)$. Under these settings, the average training error rate over the posterior is zero and $\mathrm{KL}(Q|P) = \ln \frac{1}{\mathbb{P}[V_S]}$. □

**Theorem 3** (Average holdout error of a binary classifier). *First, fix a probability measure $\mathbb{P}$ over the weight space $\Omega$ of a binary classifier. Let $S$ denote a training set and let $V_S$ denote the solution space. For a holdout set $T$ of $m$ datapoints, consider the holdout error $0 \leq \varepsilon_T(w) \leq 1$ of weight setting $w \in \Omega$, and its average over the solution space $\varepsilon_T(V_S) := \mathbb{E}_{w \sim \mathbb{P}}[\varepsilon_T(w) \,|\, w \in V_S]$. Then:*

$$\varepsilon_T(V_S) = \frac{1}{m} \sum_{(x,y) \in T} \frac{\mathbb{P}[V_{S \cup (x,-y)}]}{\mathbb{P}[V_S]}. \tag{11}$$

*Proof.* The result follows by interchanging the order of expectations, rewriting the expectation of an indicator variable as a probability and finally applying the definition of conditional probability:

$$
\begin{aligned}
\varepsilon_T(V_S) := \mathbb{E}_{w \sim \mathbb{P}}[\varepsilon_T(w) \,|\, w \in V_S] &= \frac{1}{m} \sum_{(x,y) \in T} \mathbb{E}_{w \sim \mathbb{P}}\left[\mathbb{I}[\mathrm{sign}\, f(x, w) = -y] \,|\, w \in V_S\right] \\
&= \frac{1}{m} \sum_{(x,y) \in T} \mathbb{P}\left[\mathrm{sign}\, f(x, w) = -y \,|\, w \in V_S\right] \\
&= \frac{1}{m} \sum_{(x,y) \in T} \frac{\mathbb{P}\left[\mathrm{sign}\, f(x, w) = -y \text{ and } w \in V_S\right]}{\mathbb{P}\left[w \in V_S\right]} \\
&= \frac{1}{m} \sum_{(x,y) \in T} \frac{\mathbb{P}[V_{S \cup (x,-y)}]}{\mathbb{P}[V_S]}.
\end{aligned}
$$

□

### A.2   GAUSSIAN ORTHANT PROBABILITIES

**Lemma 1** (Gaussian orthant probability). *For a covariance matrix $\Sigma \in \mathbb{R}^{n \times n}$, and a binary vector $y \in \{\pm 1\}^n$, let $p$ denote the corresponding Gaussian orthant probability:*

$$p := \mathbb{P}_{\varphi \sim \mathcal{N}(0, \Sigma)}[\mathrm{sign}(\varphi) = y]. \tag{7}$$

*Letting $\mathbb{I}$ denote the $n \times n$ identity matrix, $\odot$ the elementwise product and $|\cdot|$ the elementwise absolute value, then $p$ may be equivalently expressed as:*

$$p = \frac{1}{2^n} \mathbb{E}_{u \sim \mathcal{N}(0, \mathbb{I})} \left[ \mathrm{e}^{-\frac{1}{2}(y \odot |u|)^T \left( \sqrt[n]{\det \Sigma}\, \Sigma^{-1} - \mathbb{I} \right)(y \odot |u|)} \right] =: \mathrm{e}^{-\mathcal{C}_0(\Sigma, y)}, \tag{8}$$

*and $p$ may be bounded as follows:*

$$p \geq \frac{1}{2^n} \mathrm{e}^{\frac{n}{2} - \sqrt[n]{\det \Sigma}\left[ \left(\frac{1}{2} - \frac{1}{\pi}\right) \mathrm{tr}(\Sigma^{-1}) + \frac{1}{\pi} y^T \Sigma^{-1} y \right]} =: \frac{1}{2^n} \mathrm{e}^{\frac{n}{2} - \mathcal{C}_1(\Sigma, y)}. \tag{9}$$

*Proof.* The orthant probability may first be expressed using the probability density function of the multivariate Normal distribution as follows:

$$p = \frac{1}{\sqrt{(2\pi)^n \det \Sigma}} \int_{y \odot \varphi \geq 0} e^{-\frac{1}{2}\varphi^T \Sigma^{-1} \varphi} \, d\varphi.$$

By the change of variables $u = \frac{y \odot \varphi}{\sqrt[2n]{\det \Sigma}}$ or equivalently $\varphi = \sqrt[2n]{\det \Sigma}(y \odot u)$, the orthant probability may be expressed as:

$$
\begin{aligned}
p &= \frac{1}{\sqrt{(2\pi)^n}} \int_{u \geq 0} e^{-\frac{1}{2}(y \odot u)^T \sqrt[n]{\det \Sigma}\Sigma^{-1}(y \odot u)} \, du \\
&= \frac{1}{2^n} \frac{1}{\sqrt{(2\pi)^n}} \int_{\mathbb{R}^n} e^{-\frac{1}{2}(y \odot |u|)^T \sqrt[n]{\det \Sigma}\Sigma^{-1}(y \odot |u|)} \, du \\
&= \frac{1}{2^n} \mathbb{E}_{u \sim \mathcal{N}(0,\mathbb{I})} \left[ e^{-\frac{1}{2}(y \odot |u|)^T \left( \sqrt[n]{\det \Sigma}\Sigma^{-1} - \mathbb{I} \right)(y \odot |u|)} \right],
\end{aligned}
$$

where the second equality follows by symmetry, and the third equality follows by inserting a factor of $e^{-u^2/2}e^{+u^2/2} = 1$ into the integrand.

Next, by Jensen's inequality,

$$
\begin{aligned}
p &\geq \frac{1}{2^n} e^{-\frac{1}{2} \mathbb{E}_{u \sim \mathcal{N}(0,\mathbb{I})}\left[ (y \odot |u|)^T \left( \sqrt[n]{\det \Sigma}\Sigma^{-1} - \mathbb{I} \right)(y \odot |u|) \right]} \\
&= \frac{1}{2^n} e^{-\frac{1}{2} \sum_{ij} \mathbb{E}_{u \sim \mathcal{N}(0,\mathbb{I})}\left[ y_i y_j |u_i||u_j| \left( \sqrt[n]{\det \Sigma}\Sigma^{-1}_{ij} - \delta_{ij} \right) \right]} \\
&= \frac{1}{2^n} e^{-\frac{1}{2}\left[ \sum_i \left( \sqrt[n]{\det \Sigma}\Sigma^{-1}_{ii} - 1 \right) + \frac{2}{\pi} \sum_{i \neq j} y_i y_j \sqrt[n]{\det \Sigma}\Sigma^{-1}_{ij} \right]} \\
&= \frac{1}{2^n} e^{-\frac{1}{2}\left[ \sum_i \left( (1 - \frac{2}{\pi}) \sqrt[n]{\det \Sigma}\Sigma^{-1}_{ii} - 1 \right) + \frac{2}{\pi} \sum_{ij} y_i y_j \sqrt[n]{\det \Sigma}\Sigma^{-1}_{ij} \right]} \\
&= \frac{1}{2^n} e^{-\frac{1}{2}\left[ (1 - \frac{2}{\pi}) \sqrt[n]{\det \Sigma} \operatorname{tr}(\Sigma^{-1}) - n + \frac{2}{\pi} \sqrt[n]{\det \Sigma} y^T \Sigma^{-1} y \right]} \\
&= \frac{1}{2^n} e^{\frac{n}{2} - \sqrt[n]{\det \Sigma}\left[ (\frac{1}{2} - \frac{1}{\pi}) \operatorname{tr}(\Sigma^{-1}) + \frac{1}{\pi} y^T \Sigma^{-1} y \right]}.
\end{aligned}
$$

The third equality follows by noting $\mathbb{E}_{u \sim \mathcal{N}(0,\mathbb{I})} |u_i||u_i| = \mathbb{E}_{u \sim \mathcal{N}(0,1)} u^2 = 1$, while for $i \neq j$, $\mathbb{E}_{u \sim \mathcal{N}(0,\mathbb{I})} |u_i||u_j| = [\mathbb{E}_{u \sim \mathcal{N}(0,1)} |u|]^2 = 2/\pi$. $\qquad \square$

## A.3 Neural Networks as Gaussian Processes

The essence of the following lemma is due to Neal (1994). The lemma will be used in the proof of Theorem 4.

**Lemma 2** (NNGP correspondence). *For the neural network layer given by Equation 18, consider randomly sampling the weight matrix $W^{(l)}$. If the following hold:*

(i) *for every $x \in \mathbb{R}^{d_0}$, the activations $\varphi\left(z_1^{(l-1)}(x)\right), ..., \varphi\left(z_{d_{l-1}}^{(l-1)}(x)\right)$ are iid with finite first and second moment;*

(ii) *the weights $W_{ij}^{(l)}$ are drawn iid with zero mean and finite variance;*

(iii) *for any random variable $z$ with finite first and second moment, $\varphi(z)$ also has finite first and second moment;*

*then, in the limit that $d_{l-1} \to \infty$, the following also hold:*

(1) *for every $x \in \mathbb{R}^{d_0}$, the activations $\varphi\left(z_1^{(l)}(x)\right), ..., \varphi\left(z_{d_l}^{(l)}(x)\right)$ are iid with finite first and second moment;*

(2) *for any collection of $k$ inputs $x^{(1)}, ..., x^{(k)}$, the distribution of the $i$th pre-activations $z_i^{(l)}(x^{(1)}), ..., z_i^{(l)}(x^{(k)})$ is jointly Normal.*

While condition (i) may seem non-trivial, notice that the lemma propagates this condition to the next layer via entailment (1). This means that provided condition (i) holds for $\varphi(z^{(1)}(x))$ at the first layer, then recursive application of the lemma implies that the network's pre-activations are jointly Normal *at all layers* via entailment (2).

*Proof of Lemma 2.* To establish entailment (1), consider the $d_l$-dimensional vector $Z_1 := \left[z_1^{(l)}(x), ..., z_{d_l}^{(l)}(x)\right]$. Observe that $Z_1$ satisfies:

$$Z_1 = \frac{1}{\sqrt{d_{l-1}}} \sum_{j=1}^{d_{l-1}} \left[W_{1j}^{(l)}\varphi\left(z_j^{(l-1)}(x)\right), ..., W_{d_l j}^{(l)}\varphi\left(z_j^{(l-1)}(x)\right)\right]. \tag{15}$$

By conditions (i) and (ii), the summands in Equation 15 are iid random vectors with zero mean, and any two distinct components of the same vector summand have the same variance and zero covariance. Then by the multivariate central limit theorem (van der Vaart, 1998, p. 16), in the limit that $d_{l-1} \to \infty$, the components of $Z_1$ are Gaussian with a covariance equal to a scaled identity matrix. In particular, the components of $Z_1$ are iid with finite first and second moment. Applying condition (iii) then implies that the same holds for $\varphi(Z_1)$. This establishes entailment (1).

To establish entailment (2), consider instead the $k$-dimensional vector $Z_2 := \left[z_i^{(l)}(x^{(1)}), ..., z_i^{(l)}(x^{(k)})\right]$. Observe that $Z_2$ satisfies:

$$Z_2 = \frac{1}{\sqrt{d_{l-1}}} \sum_{j=1}^{d_{l-1}} \left[W_{ij}^{(l)}\varphi\left(z_j^{(l-1)}(x^{(1)})\right), ..., W_{ij}^{(l)}\varphi\left(z_j^{(l-1)}(x^{(k)})\right)\right]. \tag{16}$$

Again by combining conditions (i) and (ii), the summands in Equation 16 are iid random vectors with finite mean and finite covariance. Then as $d_{l-1} \to \infty$, the distribution of $Z_2$ is jointly Normal—again by the multivariate central limit theorem. This establishes entailment (2). □

The essence of the following theorem appears in a paper by Lee et al. (2018), building on the work of Cho & Saul (2009). The theorem and its proof are included for completeness.

**Theorem 4** (NNGP for relu networks)**.** *Consider an L-layer MLP defined recursively via:*

$$z_i^{(1)}(x) = \frac{1}{\sqrt{d_0}} \sum_{j=1}^{d_0} W_{ij}^{(1)} x_j, \tag{17}$$

$$z_i^{(l)}(x) = \frac{1}{\sqrt{d_{l-1}}} \sum_{j=1}^{d_{l-1}} W_{ij}^{(l)}\varphi\left(z_j^{(l-1)}(x)\right), \tag{18}$$

*where $x \in \mathbb{R}^{d_0}$ denotes an input, $z^{(l)}(x) \in \mathbb{R}^{d_l}$ denotes the pre-activations at the lth layer, $W^{(l)}$ denotes the weight matrix at the lth layer, and $\varphi$ denotes the nonlinearity.*

*Set the output dimension $d_L = 1$ and set the nonlinearity to a scaled relu $\varphi(z) := \sqrt{2} \cdot \max(0, z)$. Suppose that the weight matrices $W^{(1)}, ..., W^{(L)}$ have entries drawn iid $\mathcal{N}(0, 1)$, and consider any collection of k inputs $x^{(1)}, ..., x^{(k)}$ each with Euclidean norm $\sqrt{d_0}$.*

*If $d_1, ..., d_{L-1} \to \infty$, the distribution of outputs $z^{(L)}(x^{(1)}), ..., z^{(L)}(x^{(k)}) \in \mathbb{R}$ induced by random sampling of the weights is jointly Normal with mean zero and covariance:*

$$\mathbb{E}\left[z^{(L)}(x)z^{(L)}(x')\right] = \underbrace{h \circ ... \circ h}_{L-1 \text{ times}}\left(\frac{x^T x'}{d_0}\right); \tag{19}$$

*where $h(t) := \frac{1}{\pi}\left[\sqrt{1-t^2} + t \cdot (\pi - \arccos t)\right]$.*

*Proof.* Condition (ii) of Lemma 2 holds at all layers for iid standard Normal weights, and condition (iii) holds trivially for the scaled relu nonlinearity. Provided one can establish condition (i) for the first layer activations $\varphi\left(z_1^{(1)}(x)\right), ..., \varphi\left(z_{d_1}^{(1)}(x)\right)$, then condition (i) will hold at all layers by

recursive application of Lemma 2, thus establishing joint Normality of the pre-activations at all layers (including the network outputs). But condition (i) holds at the first layer, since it is quick to check by Equation 17 that for any $x$ satisfying $\|x\|_2 = \sqrt{d_0}$, the pre-activations $z_1^{(1)}(x), ..., z_{d_1}^{(1)}(x)$ are iid $\mathcal{N}(0, 1)$, and $\varphi$ preserves both iid-ness and finite-ness of the first and second moment.

Since the pre-activations at any layer are jointly Normal, all that remains is to compute their first and second moments. For the $i$th hidden unit in the $l$th layer, the first moment $\mathbb{E}[z_i^{(l)}(x)] = 0$. This can be seen by taking the expectation of Equation 18 and using the fact that the $W_{ij}^{(l)}$ are independent of the previous layer's activations and have mean zero.

Since the pre-activations $z_i^{(l)}(x)$ and $z_i^{(l)}(x')$ are jointly Normal with mean zero, their distribution is completely described by their covariance matrix $\Sigma_l(x, x')$, defined by:

$$\rho_l(x, x') := \mathbb{E}\left[z_i^{(l)}(x)z_i^{(l)}(x')\right]$$

$$\Sigma_l(x, x') := \begin{bmatrix} \rho_l(x, x) & \rho_l(x, x') \\ \rho_l(x, x') & \rho_l(x', x') \end{bmatrix},$$

where the hidden unit index $i$ is unimportant since hidden units in the same layer are identically distributed.

The theorem statement will follow from an effort to express $\Sigma_l(x, x')$ in terms of $\Sigma_{l-1}(x, x')$, and then recursing back through the network. By Equation 18 and independence of the $W_{ij}^{(l)}$, the covariance $\rho_l(x, x')$ may be expressed as:

$$\rho_l(x, x') = \mathbb{E}\left[\varphi\left(z_j^{(l-1)}(x)\right)\varphi\left(z_j^{(l-1)}(x')\right)\right], \tag{20}$$

where $j$ indexes an arbitrary hidden unit in the $(l-1)$th layer. To make progress, it helps to first evaluate:

$$\rho_l(x, x) = \mathbb{E}\left[\varphi\left(z_j^{(l-1)}(x)\right)^2\right] = \frac{1}{2} \cdot 2 \cdot \rho_{l-1}(x, x),$$

which follows by the definition of $\varphi$ and symmetry of the Gaussian expectation around zero. Then, by recursion:

$$\rho_l(x, x) = \rho_{l-1}(x, x) = ... = \rho_1(x, x) = 1,$$

where the final equality holds because the first layer pre-activations are iid $\mathcal{N}(0, 1)$ by Equation 17. Therefore, the covariance $\Sigma_{l-1}$ at layer $l-1$ is just:

$$\Sigma_{l-1}(x, x') = \begin{bmatrix} 1 & \rho_{l-1}(x, x') \\ \rho_{l-1}(x, x') & 1 \end{bmatrix},$$

Equation 20 may now be used to express $\rho_l(x, x')$ in terms of $\rho_{l-1}(x, x')$. Dropping the $(x, x')$ indexing for brevity:

$$\rho_l = \mathbb{E}_{u, v \sim \mathcal{N}(0, \Sigma_{l-1})}\left[\varphi(u)\varphi(v)\right]$$

$$= \frac{1}{\pi\sqrt{1 - \rho_{l-1}^2}} \iint_{u, v \geq 0} \mathrm{d}u\,\mathrm{d}v \exp\left[-\frac{u^2 - 2\rho_{l-1}uv + v^2}{2(1 - \rho_{l-1}^2)}\right] uv.$$

By making the substitution $\rho_{l-1} = \cos\theta$, this integral becomes equivalent to $\frac{1}{\pi}J_1(\theta)$ as expressed in Equation 15 of Cho & Saul (2009). Substituting in the evaluation of this integral (Cho & Saul, 2009, Equation 6), one obtains:

$$\rho_l(x, x') = h(\rho_{l-1}(x, x')), \tag{21}$$

where $h(t) := \frac{1}{\pi}\left[\sqrt{1 - t^2} + t \cdot (\pi - \arccos t)\right]$.

All that remains is to evaluate $\rho_1(x, x')$. Since $\mathbb{E}\left[W_{ij}^{(1)}W_{ik}^{(1)}\right] = \delta_{jk}$, this is given by:

$$\rho_1(x, x') := \mathbb{E}\left[z_i^{(1)}(x)z_i^{(1)}(x')\right]$$

$$= \frac{1}{d_0}\sum_{j,k=1}^{d_0}\mathbb{E}\left[W_{ij}^{(1)}W_{ik}^{(1)}\right]x_j x_k' = \frac{x^T x'}{d_0}.$$

The proof is completed by combining this expression with the recurrence relation in Equation 21. □

