# OpenReview forum: "On the Implicit Biases of Architecture & Gradient Descent"
_ICLR.cc/2022/Conference — ICLR 2022 Submitted_

### Official Review · Reviewer_CXBm · 2021-11-01

**Correctness:** 3
**Technical Novelty And Significance:** 3
**Empirical Novelty And Significance:** 2
**Recommendation:** 6
**Confidence:** 3

**Main Review:**

This paper explores an important open question in deep learning which is what factors are contributing to the impressive generalization error for deep learning even though no existing theories can adequately explain it. The paper has quite a few strong points but also some weaknesses, both of which I will discuss below.

**Pros**:
- The paper is very well written and easy to follow. The presentation of all the results and how they fit together is very clear.
- The experiments on the comparison of different widths of MLP and the infinite width limit are very interesting. It suggests that using infinite width may be a reasonable approximation for what happens at finite width.
- The discussion of margin is very insightful and it demonstrates that a pure NNGP based approach is insufficient as an explanation for generalization (even though in relatively simplistic settings).

**Cons**:
- The claim is that the bound can accurately reflect the influence of architectures but in reality, it only applies to MLPs of different depths. While this may be due to the fact that MLPs are amenable to theoretical analysis, it also precludes the results from explaining actual neural networks architectures having different generalization performances. For example, in the bottom-right of figure 2, the separation between the actual performance of depth 2 and depth 7 MLPs are not that large and are clearly vanishing as the number of data points increase (which implies for large numbers of data points the two architectures are not too different). I think the results would be much stronger if similar results can be shown for say CNN (which has intuitive inductive biases for images) or ResNet, although I concede that the analytical bounds may be highly non-trivial to derive.
- Only Nero is used for optimization, which is a quite new algorithm. What is the reasoning behind this design choice? Do the same experiments hold for other optimization methods? Furthermore, if different optimization algorithms exhibit different generalization performances, where does the phenomenon fit in the proposed framework?
- The non-vacuousness of the bound feels like a consequence of the dataset (i.e., it’s much simpler than regular tasks) rather than the consequence of the bounding technique, so I am not sure if it’s significant. For example, would this still work if you use cifar 10 or imagenet?

**Questions**:
- It is not clear to me what the significance of theorem 3 is. With access to held-out data, one could always easily compute the test error. Perhaps the importance is that this can be applied to infinitely wide neural networks?
- In practice, we know that architecture plays a huge role in how well the model generalizes, which under the paper’s framework would suggest that these architectures would naturally induce high intraclass correlation. It feels like this is something that can be empirically tested for popular architectures (e.g., VGG vs Resnet) even if the analytical bound cannot be computed. Likewise, it would indirectly suggest that as the architecture gets better, the effect of max-margin would be diminishing. This might also explain why some NNGP works have performance comparable to or even better than GD.
- In [1], the authors discuss the potential insufficiency of existing generalization bounds. In the appendix, there was a discussion about similar problems of PAC-Bayes bounds. Can you discuss how your result fits in [1]?

**Reference**:

[1] Uniform convergence may be unable to explain generalization. Nagarajan et al.

-------------

**Update**

I thank the authors for the response. My questions have been answered but I feel my concerns are not adequately addressed. First, regarding analytical solutions for CNN, the response does not really address the question which is that the technique in its current form does not apply to CNN. If expectation-propagation approximation does just as well then the value of having an analytical bound would be small. If that is the case, then the main contribution of this paper would be the discussion of margin. I have read the discussion between the authors and reviewer HuiV and agree with reviewer HuiV that the discussion of margin could use more careful treatment. This is more true if the authors believe or the experiments show that expectation-propagation is good enough for computing the bound.

Regarding Nero, I am not convinced that it is sufficient to use the observed results to generalize all of the gradient-based optimization methods. If the authors believe that it is, relevant empirical results should be included in the paper.

Due to these reasons, I intend to keep my score.



**Summary Of The Paper:**

The paper investigates the roles of architecture and gradient descent in generalization through the lens of infinitely wide neural networks (i.e., neural network gaussian process). The paper first presents an analytical unbiased estimator of the NNGP’s generalization error and shows that the generalization depends on both the training data and how well the architecture induces correlation between the data points. The approach also offers statistical consistency for the Monte Carlo estimation which previous methods do not. The paper then investigates why these bounds are still very loose compared to the actual generalization error and finds that gradient descent favors solutions with large margins and concludes that the property explains the gap between the NNGP bound and observed performance.

**Summary Of The Review:**

The paper presents some insightful results on the implicit bias on architecture and gradient descent as an optimization method. The theoretical discussion is interesting but the experimental validation could be improved.

---

> ### Author Response · Authors · 2021-11-22
> **Reply to Reviewer CXBm**
>
> Dear Reviewer CXBm,
>
> We are grateful for your review. Thank you for commenting that our experiments are “very interesting” and that our discussion of margin is “very insightful”. We also appreciate your critiques.
>
> To address your specific critiques:
>
> - “I think the results would be much stronger if similar results can be shown for say CNN”
>
> We do feel that our results are already quite interesting even without treating CNNs analytically (which would be an exciting direction for future work). That said, we do point your attention to arxiv.org/abs/2012.04115, where Valle-Pérez and Louis test function space PAC-Bayes theory for CNNs using the expectation-propagation approximation.
>
> - “Only Nero is used for optimization, which is a quite new algorithm. What is the reasoning behind this design choice? Do the same experiments hold for other optimization methods?”
>
> We chose Nero since it allowed us to run lots of experiments with varied neural architectures without needing to re-tune optimiser hyperparameters (which can become very expensive). Also, since Nero constrains weight norms, this factors out the trivial effect that scaling up a layer’s weights has on margin. That said, we did check that vanilla GD and Adam have qualitatively the same behaviour as Nero.
>
> - “Furthermore, if different optimization algorithms exhibit different generalization performances, where does the phenomenon fit in the proposed framework?”
>
> Nothing in our paper precludes small differences in generalisation between different optimisers, but as mentioned we found that the overall effect of margin does persist for vanilla gradient descent and Adam.
>
> - “The non-vacuousness of the bound feels like a consequence of the dataset”
>
> We again point your attention to arxiv.org/abs/2012.04115, where Valle-Pérez and Louis run more extensive tests of the theory using the expectation-propagation approximation. (Unlike us, they do not identify the importance of margin.)
>
> And for your questions:
>
> - “It is not clear to me what the significance of theorem 3 is. With access to held-out data, one could always easily compute the test error. Perhaps the importance is that this can be applied to infinitely wide neural networks?”
>
> The significance of Theorem 3 is that it computes the average error over *all networks* that fit the training set. Thus it assesses a global property of the function space. The naïve way to compute this is to run every possible neural network on the training set, throw away any network that does not get 100% train accuracy, then compute the average holdout error of all the networks that are left over.
>
>
> - “In practice, we know that architecture plays a huge role in how well the model generalizes, which under the paper’s framework would suggest that these architectures would naturally induce high intraclass correlation. It feels like this is something that can be empirically tested for popular architectures”
>
> Thank you for this idea, which we think is a very good one.
>
> - “Likewise, it would indirectly suggest that as the architecture gets better, the effect of max-margin would be diminishing. This might also explain why some NNGP works have performance comparable to or even better than GD.”
>
> If you look at Figure 3 top right, we see that the performance of NNGP and NN at margin 1 is actually very similar. We believe that this may be why many works find similar performance between NNGP and NN.
>
> - “In [1], the authors discuss the potential insufficiency of existing generalization bounds. In the appendix, there was a discussion about similar problems of PAC-Bayes bounds. Can you discuss how your result fits in [1]?”
>
> This is a great question. The paper [1] is about a type of generalisation theory called “uniform convergence” which derives statements about the generalisation of *all functions* in the function class, but these bounds are usually vacuous for NN. The type of PAC-Bayes bound considered in Appendix J of [1] is also a special subclass of PAC-Bayes that derives “deterministic” or “derandomised” bounds on the behaviour of individual classifiers. But as far as we are aware, this subclass of PAC-Bayes bound is also usually vacuous for NN.
>
>
> In contrast, our paper derives PAC-Bayes bounds of a more standard form. In particular, our bounds are not “deterministic”, “derandomised” or “uniform convergence”. Instead of applying to *all* classifiers or to *particular* classifiers, our bounds hold *on average* over all classifiers. One way to think about this is that our bounds govern the performance of *typical* classifiers, while allowing for the existence of classifiers that overfit so long as they occur with low probability in the function space. This is what allows our bounds to be non-vacuous.
>
> To summarise, our results are entirely consistent with the high level message of [1], since our bounds do not attempt to guarantee uniform convergence.

---

### Official Review · Reviewer_Lxe3 · 2021-11-02

**Correctness:** 3
**Technical Novelty And Significance:** 2
**Empirical Novelty And Significance:** 2
**Recommendation:** 5
**Confidence:** 3

**Main Review:**

Strong points

The provided PAC-Bayes bound is non-vacuous, which is something that is remarkable in this context.
The PAC-Bayes bound takes into account the architecture of the network, and the empirical evaluation validates how this proposal is able to capture this.
Theorem 3 is an interesting result, although not sure if this is something useful outside this specific problem.
The analysis of GD is interesting and provides some novel insights.



Weak Point:

The PAC-Bayesian bound does not directly upper bounds the generalization error of a given neural network with null training error. It is an expectation over the generalization error of neural networks with null training error.

The PAC-Bayesian bound directly builds on previous results (Valle-Pérez et al. (2019)). The contribution is mainly on the improvement of the computation of the bound. Even though, a version of the presented bound also needs of approximate (Monte Carlo)  computations. Why are not you comparing with the previous bound presented in (Valle-Pérez et al. (2019)?


When evaluating the difference between the provided bound and the results using GD, authors attribute the difference to GD: “the significant gap between the gradient descent holdout error and the holdout error averaged over solutions suggests slackness in the bounding technique in Theorem 2 may also not be the main culprit.” This is a bit confusing statement. To what extent can we be sure that the slackness of the bound is not a relevant reason for this gap?


The analysis of the implicit bias of GD looks a bit arbitrary. Recent literature (Smith et al, 2021, Barret et al. 2021) offers much more solid explanations about this implicit biased. Which are the advantages of your analysis of GD? Authors do not discuss how their contributions relate to these existing works.


References:

David G.T. Barrett, Benoit Dherin (2021) Implicit Gradient Regularization

Samuel L Smith, Benoit Dherin, David Barrett, Soham De (2021) On the Origin of Implicit Regularization in Stochastic Gradient Descent


**Summary Of The Paper:**

This work starts by introducing a new PAC-Bayes bound over the generalization performance of a neural network that takes into account the architecture of the model. The provided bound is analytically tractable and provides non-vacuous high-probability bounds.  However, when evaluated, authors note that this bound is not tight, and proceed to evaluate the reasons for this gap. According to this work, the main reason for this gap is the implicit biased introduced by the gradient descent algorithm. Then, authors show how the implicit biased of SGD is directly related to its ability to control the margin.

**Summary Of The Review:**

I acknowledge that this work introduces some relevant contributions, but, based on the above weak points,  I lean to reject this paper. I find that the main contribution (Theorem 2) to be incremental and that the rest of the analysis do not provide novel and relevant insights.

---

> ### Author Response · Authors · 2021-11-22
> **Reply to Reviewer Lxe3**
>
> Dear Reviewer Lxe3,
>
> Thank you for your review. We appreciate your comment that our non-vacuous analytical bound is “remarkable”, and that our analysis of gradient descent provides “novel insights”. We also appreciate your critiques.
>
> To address your specific questions and critiques:
>
> - “The PAC-Bayesian bound does not directly upper bound the generalization error of a given neural network with null training error. It is an expectation over the generalization error of neural networks with null training error.”
>
> As far as we are aware, there is no known generalisation bound that is both non-vacuous and upper bounds the generalization error of a given neural network with null training error. Achieving this would constitute a major breakthrough in the generalisation theory of deep learning.  To the best of our knowledge, our bound is the first non-vacuous generalisation bound for deep learning that depends analytically on details of architecture and dataset.
>
> - “Why are not you comparing with the previous bound presented in (Valle-Pérez et al. (2019)?”
>
> We believe there is an intrinsic value in the precise mathematical results that we have derived. Rather than focusing our effort on retrospectively investigating the validity of approximations used in prior work, we found it more interesting to look forward and focus on using our new tools to derive novel insights about the generalisation properties of the NN function space, including the dependence on details like margin.
>
> - “To what extent can we be sure that the slackness of the bound is not a relevant reason for this gap?”
>
> We show that a large gap exists even with a direct estimate of the bounded quantity. This means that even if the bound was perfectly tight, a large gap would still exist.
>
> - “The analysis of the implicit bias of GD looks a bit arbitrary… Authors do not discuss how their contributions relate to… existing works.”
>
> We do not agree that our analysis is arbitrary. We point out a simple relationship between margin and generalisation that exists in theory for GP classification (Equations 12 and 13) and in practice for both GP classification and deep learning (Figure 3, top right). Neither of the works mentioned by the reviewer point out this effect.

---

> > ### Comment · Reviewer_Lxe3 · 2021-11-29
> > **Premature paper**
> >
> > Dear Authors,
> >
> > Thanks for your response. After reading the rest of the reviews and your response, I plan to keep my score. In my opinion,
> > this paper is premature in its current form to be published in this top-tier venue. Although I can appreciate novel theoretical contributions, they are not significant enough (e.g. they do not apply to convolutional neural networks as pointed out by Reviewer CXBm), they are relatively incremental when they are compared to previously published results (Valle-Pérez et al. (2019)) and the experimental evaluation is not complete (e.g. no comparison is performed against (Valle-Pérez et al. (2019))).

---

> > > ### Author Response · Authors · 2021-12-08
> > > **Thank you for your review**
> > >
> > > Dear Reviewer Lxe3,
> > >
> > > Thank you again for your review and your feedback, which will be useful in improving our paper!

---

### Official Review · Reviewer_WQvV · 2021-11-03

**Correctness:** 3
**Technical Novelty And Significance:** 3
**Empirical Novelty And Significance:** 2
**Recommendation:** 6
**Confidence:** 2

**Main Review:**

## Strengths

1. The main finding of the paper, namely the huge disparity between the GD NN and NNGP in section 5.2 is novel, and thought-provoking.

1. The paper is clearly and carefully written, and I notably appreciate the authors injecting notes to provide intuition for their bounds.

1. The question of disentangling architecture from the training procedure in the context of generalization is an active and important area of research to which this paper is relevant.

## Weaknesses

1. My main concern with the paper is that, based on presented experiments, I believe the central claim about GD providing a boost to generalization through large margin-seeking behavior (section 5) is likely very specific to the precise setting considered in the paper. Namely:

	1. The authors use a very recent "Nero'' optimizer (https://arxiv.org/abs/2102.07227) instead of GD or even SGD. From skimming the paper "Nero'' appears to substantially differ from (S)GD, and provide generalization benefits itself over other optimizers. The authors also mention its importance after Equation (14) since it constrains the norms of weights. Further, but on a minor note, the authors use mini-batching, and learning rate (LR) decay. Overall, this setting is very far from what I consider to be vanilla GD, and not adequate to draw conclusions about GD in general.
    	* I think to properly discuss the inductive bias of GD, the authors should first clearly define what they mean by GD, and then ablate against different settings that lie between NNGP and GD. For example, GD could be defined as infinite-width, full-batch gradient flow, which yields the Neural Tangent Kernel (https://arxiv.org/abs/1806.07572; NTK), in which case one would only need to compare NNGP and NTK. Alternatively, one could consider finite-width, full-batch, gradient flow, in which case it would be best to compare three settings of NNGP, NTK, and finite-width, full-batch, small LR GD NN. If one defines GD to be simply full-batch, finite-width, then experiments should contain NNGP, NTK, and a set of results for full-batch, finite-width GD for various LRs, since a specific LR is not defined to be a part of GD. If GD stands for just any first-order optimizer, then other optimizers like GD, SGD, Adam etc. should be considered along with “Nero”. And so on. In other words, the point of this tedious comment is that there are many conceptual steps between NNGP and "Nero'' (infinite Bayesian (NNGP) -> infinite gradient flow (NTK) -> finite gradient flow -> finite GD -> finite SGD -> finite SGD with a specific learning rate -> ... -> "Nero" with a specific LR, LR schedule, weight normalization, and a mini-batch size), and by looking only at the extremes of this spectrum (NNGP and "Nero") it is hard to understand what aspect of "Nero" is responsible for the observed effect. For instance, perhaps it's all about constraining the norms of the weights as mentioned after Equation (14), and perhaps vanilla GD suffers from the same issues as NNGP.

    1. Even "Nero" optimizer aside, I find presented evidence quite limited:

    	1. Section 5.3 indeed shows a setting where NNGP will fail for large $\alpha$, but I find it quite artificial. One could easily define inputs/outputs rescaling as part of the model, and it is in fact an extremely common practice in ML to normalize the data to lie within appropriate ranges. So any effect observed as a consequence of simple data rescaling is not very significant in my opinion, notably when one of the methods ("Nero") does weight norm-constraining. Perhaps one way to improve this is to consider datasets where different labels $y_i$ are multiplied by different $\alpha_i$, resulting in no change in scale on average (not sure at the top of my head what would be the fair setting i.e. $\sum_i \alpha_i = 1$, or $\sum_i \alpha_i^2 = 1$, or maybe even something dependent on $\Sigma_{SS}$), while still giving opportunity for margin-seeking methods to show benefits.

		1. Section 5.2 does show a very strong advantage of GD ("Nero") over NNGP in an apparently realistic setting. However, this result is contrary to quite a bit of existing literature (e.g. https://arxiv.org/abs/1711.00165,
https://arxiv.org/abs/1810.05148,
https://arxiv.org/abs/1910.01663,
https://arxiv.org/abs/2007.15801) claiming that MLP NNGP / NTK perform on par or better than GD MLPs, especially on small datasets. For this reason I believe this kind of claim requires more substantial evidence, i.e. evaluation over different datasets of different sizes, different architectures, etc. Further, given my argument about section 5.3 above, I would appreciate experimental details / evidence showing that the performance disparity _cannot_ be eliminated by simply rescaling the data, and, as above, that it's indeed the disparity between NNGP and GD and not e.g. between SGD and "Nero".

	1. Please note that preferably, in all claims regarding NNGP generalization relative to GD, for fair comparison the strength $\lambda$ of the matrix inversion regularizer, i.e. $\left(\Sigma_{SS} + \lambda I\right)^{-1}$ should be tuned, since it is an important hyper-parameter for NNGP/NTK methods (see e.g. Figure 7 in https://arxiv.org/pdf/2007.15801.pdf).


1. IIUC Equation (11) still requires Monte Carlo sampling - could you elaborate on how this compares to simply sampling from the NNGP posterior on the test set and computing the average error over the samples? Is your formula computationally cheaper, does the estimator converge faster, etc? Since it is mentioned as one of the core contributions, it would be good to know how your approach is better.


## Other

1. When citing NNGP works, please also cite https://arxiv.org/abs/1804.11271, as this work was concurrent with https://arxiv.org/abs/1711.00165.


**Summary Of The Paper:**

There is a tension in existing literature regarding the relative importance of (a) architecture and (b) gradient descent (GD) for neural network (NN) performance. This paper presents empirical evidence and theoretical reasoning in favor of GD providing a substantial benefit to generalization on top of the architecture bias. Precisely, the authors demonstrate a setting in which  finite, GD-trained NNs substantially outperform Bayesian finite and infinite-width models of the same architecture.

The authors also propose two bounds on the expected classification error of a Gaussian process (precisely, infinite-width Bayesian NNs, "neural network-Gaussian process"; NNGP), and a way to estimate the expected classification error of the NNGP. There is a big gap between the bounds and empirical performance of wide GD-trained NNs, and the authors conjecture that the gap is due to the beneficial "large margin-seeking" bias of GD, enabling superior performance of GD NNs over NNGPs mentioned above.


**Summary Of The Review:**

# Post-rebuttal update

I thank the authors for correctly pointing out that I misunderstood their work I my initial review. Given their correction, I see this as a stronger submission, and raise my score. However, I still find it lacking comprehensive experiments to claim that their bound/estimator reflect the implicit bias of the architecture, and the discussion about margin is either lacking rigor or is not completely clear to me at this time (namely, I see why for NNGP concentration on the mean is beneficial, but I struggle to understand the precise analogy with the GD, given that NTK posterior mean can often underperform compared to GD networks).

Given that I post this update late in the discussion period, the paper contains otherwise decent ideas, and I may be misunderstanding some aspects of the margin discussion, I am willing to score this as a weak accept but with low confidence.

# Original review

While I think the paper investigates interesting and important ideas, and the writing appears of high quality, I find that experimental evidence for the claim that GD provides a substantial benefit beyond NNGP (section 5.2 and 5.3) is limited to a highly specific setting. I see this as the  the main theme of the paper, and believe it requires much better evidence in terms of both quality ("Nero"/SGD/GD/NTK/... ablations, NNGP regularization tuning, controlling for trivial settings like dataset rescaling) and preferably quantity (different datasets, sizes, architectures etc).

Otherwise, the paper also proposes a formula for computing the average classification error of a GP in section 5.1, but I am not sure how this is better than just Monte Carlo sampling the error (perhaps the authors could address this in the rebuttal, although my main concern remains the lacking experiments above).

Finally the authors also propose two bounds on the average GP classification error. However, these bounds aren’t compared to NNGP performance (instead, finite-width GD NNs are used). If NNGP performance is similar to GD NNs in those settings or worse, then IIUC these bounds are not tight and do not appear very practical. Since I’m not an expert in this area, I am also open to being convinced of their importance in the rebuttal, however, I again emphasize that my main concern is the limited experimental setting.

Together, I commend the three efforts above, but I do not find them significant enough for publication at this time.

---

> ### Author Response · Authors · 2021-11-22
> **Reply to Reviewer WQvV**
>
> Dear Reviewer WQvV,
>
> We are grateful for your review. We first want to point out an important misunderstanding. While this misunderstanding certainly stems from our exposition, we hope that in light of our clarification you will be able to adjust your review appropriately.
>
> In particular, we do not agree with the statements that:
>
> - “The main theme of the paper” is that “GD provides a substantial benefit beyond NNGP”
> - “Section 5.2 does show a very strong advantage of GD ("Nero") over NNGP in an apparently realistic setting. However, this result is contrary to quite a bit of existing literature”
> - “Section 5.3 indeed shows a setting where NNGP will fail for large α”
>
>
> Our paper means to point out that **for both NNGP and neural networks, large margin functions generalise best**. Please see Figure 3 top right for evidence of this. We believe that the source of this misunderstanding is that in Section 5.2, we are computing the average performance of the NNGP **over all solutions**. This differs to what is usually dealt with in the NNGP literature which is the average performance of **solutions conditioned on fixed outputs**. So we believe that **our results are complementary to the existing literature, and not contrary**.
>
> Now to address the critiques raised in your review:
>
> - “The authors use a very recent "Nero'' optimizer” which is different to “vanilla GD”
>
> The primary reason we used Nero is that it let us run lots of experiments on varying neural architectures without needing to re-tune hyperparameters. We did check that vanilla GD and Adam exhibit the same behaviour as Nero after taking care to properly tune their learning rates.
>
> - “the authors should first clearly define what they mean by GD”
>
> We use gradient descent to refer generically to first order optimisation. We would refer to “vanilla GD” as “steepest descent on a Euclidean parameter space” while Nero is “first order optimisation accounting for structural properties of the NN parameter space”.
>
> - “the authors should… ablate against different settings that lie between NNGP and GD” in order to see “what aspect of "Nero" is responsible for the observed effect”
>
> Since we are pointing out that the NNGP function space and the GD-trained NN function space behave very similarly, we do not believe these experiments are necessary.
>
> - “GD could be defined as infinite-width, full-batch gradient flow, which yields the Neural Tangent Kernel”
>
> Interestingly, the effect that we point out is also present under NTK theory. To see this, inspect equation 16 in arxiv.org/abs/1902.06720. Notice that scaling up the regression targets Y scales up the NTK predictive mean, and thus effectively denoises the NTK predictions. So even under NTK theory, large margin functions are expected to generalise best since they are less random.
>
> - “perhaps it's all about constraining the norms of the weights”
>
> To see that this is not the case, observe that there is a simple dependence of margin on weight norms for relu networks. In particular, because relu networks are positive homogeneous functions, scaling up the weights at a certain layer scales up the margin by the same factor. But this dependence is “trivial” in the sense that it does not affect classification decisions, so it makes sense to “factor it out” by using an optimiser that constrains weight norms. An alternative would be to report a normalised margin equal to the margin divided by the product of weight norms.
>
> - “IIUC Equation (11) still requires Monte Carlo sampling - could you elaborate on how this compares to simply sampling from the NNGP posterior on the test set and computing the average error over the samples?”
>
> The idea here is that Equation 11 estimates the average holdout error of all solutions that get 100% training accuracy. This is a different posterior than is traditionally considered in the NNGP literature, and sampling directly from it is non-trivial.
>
> - “When citing NNGP works, please also cite https://arxiv.org/abs/1804.11271 ”
>
> Thank you for pointing out this missing citation, which we will add to the paper.
>
> - the “bounds aren’t compared to NNGP performance (instead, finite-width GD NNs are used). If NNGP performance is similar to GD NNs in those settings or worse, then IIUC these bounds are not tight and do not appear very practical.”
>
> For our purposes, this issue was addressed in the experiments in Figure 3 top right. The orange line in that plot is an estimate of the quantity bounded in Theorem 2, and we compare this to the empirical performance of both NNGP and GD NNs. While this is not investigating the slackness of the bound in Theorem 2, it is asking the more interesting question for us which is “even if Theorem 2 were perfectly tight, how would it compare to the performance of large margin GP draws or large margin GD NNs.”

---

> > ### Comment · Reviewer_WQvV · 2021-11-30
> > **Thank you for correction; raising my score, but still can't advocate strongly for acceptance.**
> >
> > Thank you for your important corrections, and apologies for misunderstanding your paper in the first place.
> >
> > You are correct that my initial criticism was misplaced. I am increasing my score to reflect that, as I won't be opposed if this paper is accepted. However due to other aspects below I still can't strongly advocate for it:
> >
> > 1. Presentation-wise, I find that the implicit bias of GD that you discuss would be more appropriately described as the implicit bias of overfitting, i.e. achieving zero training loss. As you point out, it's not specific to GD, so it was a bit confusing that you chose to attribute overfitting to GD (AFAIK implicit bias of GD usually refers to the gap between NNGP/GD in Figure 3, upper right). Respectively, discussion and comparison with a regression setting (i.e. considering models achieving 0 loss as the solution space) would be appreciated.
> >
> > 1. When presenting the bound and the estimator used to assess the implicit bias of the architecture, it would be nice to see it by actually comparing different architectures, e.g. linear kernel, RBF, NNGP, CNNGP, etc. Presented comparison between depth 2 and 7 is promising but very preliminary. I see that this has been raised by other reviewers as well and you have referred them to prior work, but I still find that lacking both explicit comparison of different architectures, and explicit comparison to prior works makes it difficult to judge the significance.
> >
> > 1. Regarding margin, to my understanding you are arguing that for NNGP, concentration on the mean is beneficial, since posterior fluctuations are random. When transferring this analogy to GD, what are you suggesting that GD concentrate on? If this is the NTK posterior mean, then this analogy may be problematic, since for example for CNNs with pooling (unlike MLPs), NTK posterior mean (e.g. https://arxiv.org/pdf/1904.11955.pdf, https://arxiv.org/pdf/2007.15801.pdf) actually underperforms compared to GD networks. Therefore I wish this section was more formally presented, otherwise it's hard to understand the exact claim and implications, and reconcile them with prior literature.
> >
> > For the reasons above my opinion on this paper remains borderline, and I generally agree with Lxe3 that it appears premature. Given my initial mistake and late reply now, I am willing to put it marginally above the threshold, but with low confidence.

---

> > > ### Author Response · Authors · 2021-12-08
> > > **Thank you for your review**
> > >
> > > Dear Reviewer WQvV,
> > >
> > > Thank you for updating your review. We are grateful for the additional feedback.
> > >
> > > We are posting some replies here for completeness:
> > > - We agree with you that the language surrounding the "implicit bias of gradient descent" might be misleading to some, as it might lead some readers to expect us to analyse the trajectory of gradient descent. Really the paper is pointing out the difference in generalisation behaviour between different regions of the function space, and then pointing out how *gradient descent + loss function* is the "vehicle" through which these different regions can be targeted. In any future draft of this work, including if this paper is accepted, we plan to revise that use of language for the sake of clarity.
> > > - *"When transferring this analogy to GD, what are you suggesting that GD concentrate on?"* This is a great question. What we intended to point out in the paper is two things: 1. under the NNGP prior on functions, large margin functions are expected to generalise better for the reasons mentioned. 2. this provides a strong motivation to use gradient descent to target this "large margin" region of the function space for finite width networks. In the rebuttal, we point out that NTK theory suggests exactly the same thing. So in summary: under two models of neural networks (both NNGP and NTK), large margin functions are expected to generalise better. Now of course these models have their respective limitations, so the crispest statement that we can make is something like: *insofar as finite width NNs trained by GD bear any degree of similarity to either NNGP or NTK, one should expect that large margin finite width NNs should generalise better.* Then we find in our experiments that this is the case.

---

### Official Review · Reviewer_HuiV · 2021-11-03

**Correctness:** 3
**Technical Novelty And Significance:** 2
**Empirical Novelty And Significance:** 1
**Recommendation:** 5
**Confidence:** 3

**Main Review:**

The paper is interesting and well written. Results seem mostly novel and technically correct.

The fusion of the different implicit biases brought forth by neural architecture and optimization is an interesting topic. In my opinion the paper over promises analysis of both, while providing only analysis of implicit bias of neural architecture under a Gaussian process and asserting that GD maximizes margin without theoretical support and with limited experimental evidence.

Strengths:
- Well written and motivated.
- Elegant theoretical results on the average error of the solution space.
- Combining landscape approach for architectural implicit bias along with trajectory induced implicit bias by GD is very interesting.

Weaknesses:
- The bounds obtained by the theoretical derivation are only slightly better than chance and there is a very significant gap from the empirical results observed.
- The analysis of the implicit bias of gradient descent due to margin is confusing and not very convincing. The improvement over the holdout set as a function of increasing margin constant $\alpha$ is not clear, how does this explain the inductive bias? This seems as a suggestion to improve the margin via this constraint rather than an explanation to what happen when using GD in the discussed setup.


Additional comments:
- Related work does not cover relevant works such as Neural Tangent Kernels and many trajectory based analyses of GD.
Minor
- Theorem 2 - why repeat the assumptions instead of referring to the assumptions of theorem 1?
- Figure 3, bottom right plot is missing a legend.


**Summary Of The Paper:**

This paper studies the implicit bias of neural architectures using a Bayesian approach.
The paper provides bounds on the average population error of infinite width neural networks that fit the training set under a Gaussian process.
The paper also discusses the role of gradient descent in the implicit bias, suggesting it controls the margin of the returned network from the solution space.

**Summary Of The Review:**

Interesting paper which suffers slightly from overpromise.
Landscape analysis is insightful but the gradient descent analysis is limited and does not explain the gap between architectural error bounds and actual error obtained.

---

> ### Author Response · Authors · 2021-11-22
> **Reply to Reviewer HuiV**
>
> Dear Reviewer HuiV,
>
> We are grateful for your review. We appreciate your comments that our paper is “interesting and well written” and contains “elegant theoretical results”. To address your specific critiques:
>
> - “The bounds obtained by the theoretical derivation are only slightly better than chance and there is a very significant gap from the empirical results observed.”
>
> This issue is the focus of section 5 in our paper, therefore we regard this as a contribution and not a weakness. As mentioned in the paper, the bound is on generalisation error averaged over the entire version space, but we found both numerical and theoretical evidence that large margin networks and large margin NNGP posterior samples both significantly outperform the version space average.
>
> Also note that even obtaining non-vacuous bounds is an achievement. Reviewer Lxe3 called it “remarkable”.
>
> - “asserting that GD maximizes margin without theoretical support and with limited experimental evidence.”
>
> There are a couple comments here. First, we show in section 5.3 how to design a loss function that causes gradient descent to target functions of a fixed margin (Equation 14).
>
> Second, although we do not emphasise this in the paper, it is well known in the literature that cross entropy loss causes gradient descent to target functions of maximum margin. For example, see section 4 of this paper:
>
> *Rosset, Zhu & Hastie, Margin Maximizing Loss Functions, NeurIPS 2003*
>
> Therefore we regard the statement that gradient descent can control or maximise margin as uncontroversial.
>
> - “The improvement over the holdout set as a function of increasing margin constant is not clear, how does this explain the inductive bias?”
>
> We point out that for Gaussian processes, holdout predictions are de-noised by increasing margin. More formally, Gaussian process samples concentrate on their mean in the limit that margin → infinity (Equations 12 and 13). We demonstrate empirically that the neural network function space displays this same qualitative behaviour. Notice that in Figure 3 top right, the holdout error both improves and concentrates as the margin is increased.

---

> > ### Comment · Reviewer_HuiV · 2021-11-24
> > **Response to rebuttal**
> >
> > Thank you for addressing the points raised.
> >
> > I answer the 3 main points of the rebuttal:
> > - I acknowledge the theoretical contribution of the paper and I believe it's a result justifying publication if the paper was better framed along these lines along with a more extensive empirical investigation.
> >
> > - Regarding section 5. I do not agree with the logic that GD has been shown to maximize margin in some settings so it is okay to make this assumption. Specifically, known results [1,2] for maximizing margin are mostly limited to exponential loss functions whereas the authors use the squared loss in section 5.3.
> >
> > - This section is still misleading, there is no implicit bias described here. The claim that increasing $\alpha$ improves generalization is clear. How does this lead to the claim "This suggests that gradient descent possesses a significant implicit bias in its ability to control margin..."?
> >
> > [1] GRADIENT DESCENT MAXIMIZES THE MARGIN OF HOMOGENEOUS NEURAL NETWORKS - Lyu, Li 2020
> > [2] The Implicit Bias of Gradient Descent on Separable Data - Soudry et al., 2018

---

> > > ### Author Response · Authors · 2021-11-24
> > > **Reply to response to rebuttal**
> > >
> > > Dear Reviewer HuiV,
> > >
> > > Thank you for your reply. We are grateful for your engagement with our rebuttal. In response to your comments:
> > >
> > > - "more extensive empirical investigation" would be good
> > >
> > > We point your attention to arxiv.org/abs/2012.04115, where Valle-Pérez and Louis provide more thorough experimental tests of the function space PAC-Bayes theory (using approximations). Since our findings with our analytical theory corroborated their results, we opted not to spend too much time (and expensive computation) going in this direction.
> > >
> > > **That said, we really feel that our findings about margin are both novel and important. We ask that you take another look at section 5 of our paper, but keeping in mind the following:**
> > >
> > > - "known results [1,2] for maximizing margin are mostly limited to exponential loss functions"
> > >
> > > We are not meaning to claim that gradient descent with any loss function will maximise margin. What we intend to claim is that:
> > >
> > > - gradient descent with exponential loss functions targets margin ∞;
> > > - gradient descent with square loss function targets margin 1;
> > > - gradient descent with α-scaled square loss (equation 14) targets margin α.
> > >
> > > This means that **gradient descent with α-scaled square loss provides a way to take explicit control over margin**. This allows us to directly test the effect of margin on generalisation. We regard this as "good experimental design"---**we are turning something that is usually left as an "implicit bias" into an explicit one that is under our control**.
> > >
> > > If you still feel that this is "misleading", perhaps we can clarify the language in section 5?

---

> > > > ### Comment · Reviewer_HuiV · 2021-11-27
> > > > **Additional comments**
> > > >
> > > > **Empirical evaluation** - If this is the case the paper should be written accordingly. Currently this is not clear when reading the paper and the reader is left with a not very convincing experimental setup.
> > > >
> > > > **Margin** - What are these statements based on? Are the authors referring to hinge loss? I don't think there is a consensus about 'GD with exponential loss functions targets margin ∞;' - this has been shown in several settings with strong assumptions on architecture and data.
> > > >
> > > > With regards to terminology, I would say that adding a parameter to the loss function would be better described as a form of regularization, but the more important thing here is to be precise regarding the margin claims which at the moment are not rigorous and raise doubts on the validity of the entire submission.

---

> > > > > ### Author Response · Authors · 2021-11-27
> > > > > **Reply to additional comments**
> > > > >
> > > > > Dear Reviewer HuiV,
> > > > >
> > > > > Thank you for continuing the discussion. With regard to terminology and writing, we will clarify these matters in the paper. But we would like to emphasise our belief that at the core of section 5 there is a novel observation about the connection between loss function, margin and generalisation based on the behaviour of Gaussian processes.
> > > > >
> > > > > - *"Margin - What are these statements based on?"*
> > > > >
> > > > > The statements about margin are based on inspecting each loss function and asking *what network outputs are necessary to minimise the loss function?*
> > > > >
> > > > > For example:
> > > > > - to minimise **softmax cross-entropy loss** (in multi class problems) the network must output +∞ on the correct class and a finite number on the incorrect classes. So **to minimise this loss, the margin must be ∞**. *[Since a network with constrained weight norms cannot output ∞, in practice cross entropy loss is only "targeting" a margin of ∞, although this value cannot actually be attained.]*
> > > > > - to minimise **α-scaled square loss** (in binary classification), the network must output ±α on each training point. So **to minimise this loss, the margin must be exactly α**. *[In our experiments with α-scaled square loss, we did find that the final trained networks had a margin of α.]*
> > > > >
> > > > > Does this clarify your question about the margin claims?

---

> > > > > > ### Comment · Reviewer_HuiV · 2021-11-30
> > > > > > **final remarks**
> > > > > >
> > > > > > Dear authors,
> > > > > > Thank you for your clarifications.
> > > > > >
> > > > > > Regarding margin - please note that results on margin for cross entropy loss mostly discuss the normalized margin (as the norm trivially tends to $\infty$) and the loss zeros only asymptotically. The result here is simplistic in the sense that in order to minimize the loss, the model has to output $\alpha y$ and this solution has margin $\alpha$, this has nothing to do with implicit bias.
> > > > > >
> > > > > > To conclude:
> > > > > > I think the paper contains interesting theoretical results (on average error of NNGP). However, the experiments show the bounds are very loose and do not have much practical significance. In addition, I think that section 5 does not present any contribution and should be removed.
> > > > > > At this point I prefer to preserve my score of weak reject. I do believe that more concise writing, focusing only on the implicit bias of NNGP, with more comprehensive experiments will be a good submission worthy of publication in ICLR (or similar top-tier conferences).

---

> > > > > > > ### Author Response · Authors · 2021-12-01
> > > > > > > **Final remarks from us**
> > > > > > >
> > > > > > > Dear Reviewer HuiV,
> > > > > > >
> > > > > > > We are grateful for your feedback on how to improve the paper, in terms of honing and focusing the results.
> > > > > > >
> > > > > > > A few final remarks from us:
> > > > > > >
> > > > > > > - we use an optimiser that constrains weight norms, so margin and normalized margin are equivalent in our setup.
> > > > > > > - we appreciate your point that since we show how to explicitly control (normalized) margin, it is no longer an implicit bias for us.
> > > > > > > - we believe that in deep learning, deriving an analytical generalisation bound that is non-vacuous even for a few hundred datapoints is already an accomplishment. Deriving a tighter function space PAC-Bayes bound is an interesting challenge. Our results suggest that the key may be to incorporate margin into the theory.
> > > > > > >
> > > > > > > Thank you again for reviewing our work!

---

### Author Response · Authors · 2021-11-22
**Comment to AC and all reviewers**

Dear Area Chair and Reviewers,

We are grateful for your time and effort in this process.

**Important clarification**

First, we want to clarify an important misunderstanding by one of the reviewers about the results and claims of the paper. Reviewer WQvV interpreted our paper as saying that “GD provides a substantial benefit beyond NNGP” which is “contrary to quite a bit of existing literature”. Based on this, Reviewer WQvV asked for an extensive chain of experiments to isolate the source of this difference. But the claim of our paper is actually that **for both NNGP and GD-trained neural nets, large margin functions generalise better**. This claim is compatible with and complements the existing literature on NNGP, and is supported by our results in Figure 3, top right.

The source of this misunderstanding is a confusion about what is being reported in section 5.2. In particular, our bounds and estimators (Theorems 2 and 3) are on the average error over the entire solution space:

- $\mathbb{E}_{w\sim\mathbb{P}} [\epsilon_D(w) | \mathrm{sign} f(x_1,w) = y_1,..., \mathrm{sign} f(x_n,w) = y_n].$

This is very different to what is usually considered in the NNGP literature, which is the average error conditioned on fixed outputs:

- $\mathbb{E}_{w\sim\mathbb{P}} [\epsilon_D(w) | f(x_1,w) = y_1,..., f(x_n,w) = y_n].$

**Why we think our results are exciting**

As reviewer WQvV points out, there is “a tension in existing literature regarding the relative importance of (a) architecture and (b) gradient descent (GD) for neural network (NN) performance”, and this is an “important area of research”.

We derive, to the best of our knowledge, the first non-vacuous generalisation bounds for neural nets with an analytical dependence on properties of the neural architecture and training data. Reviewer HuiV calls these results “elegant”, while reviewer Lxe3 finds it “remarkable” that our bounds are non-vacuous.

We go on to show that our bounds are insufficient to explain the full performance of GD trained neural nets *because the bounds average over functions of all margin*. In particular we show that **for both GP classification (in theory and practice) and NN classification (in practice) large margin functions generalise better**.

This final result relies on a crisp theoretical connection between large margin GP draws and the GP posterior mean. In particular, **conditioned on large outputs, the entire GP posterior concentrates on its mean**. This suggests large margin functions should tend to generalise better since they tend to contain less random fluctuation.

Since the GP posterior mean is equivalent to kernel regression (https://arxiv.org/abs/1807.02582), this suggests a new exciting connection: **large margin GP posterior samples are equivalent to min-RKHS-norm kernel regression**. Insofar as the NN function space is similar to a GP, this also suggests that **large margin NNs are roughly min-RKHS-norm kernel classifiers**.

---

### Decision · Program_Chairs · 2022-01-20

**Decision:**

Reject

**Comment:**

This is an interesting paper aiming to further advance the knowledge of implicit bias in deep networks.  Unfortunately, the reviewers had many concerns about technical details and presentation.  One concern was about section 5, on margins and implicit bias.  Oddly, this section 5 does not cite the extensive literature on margin maximization, implicit bias, and implicit regularization in deep learning (despite a mention of Soudry et al earlier on), whereas the choice of paper title and also this section title would suggest an advance over this work, or at least reference to this work (which goes far beyond that one paper); instead, that section left me a bit confused about the suggested bias and its implications on generalization.  As such, I suggest the authors spend more time on their submission, aiming to further separate their work from prior work, and address the comments of reviewers.